



# SICNet_season V1.0: a transformer-based deep learning model for seasonal Arctic sea ice prediction by incorporating sea ice thickness data

Yibin Ren[1], Xiaofeng Li[1], Yunhe Wang[1]

[1]Key Laboratory of Ocean Observation and Forecasting, Key Laboratory of Ocean Circulation and Waves, Institute of Oceanology, Chinese Academy of Sciences, Qingdao, China

*Correspondence to*: Xiaofeng Li (lixf@qdio.ac.cn)

**Abstract.** The Arctic sea ice is suffering dramatic retreating in summer and fall, which has far-reaching consequences on the global climate and commercial activities. Accurate seasonal sea ice predictions are significant in inferring climate change and planning commercial activities. However, seasonally predicting the summer sea ice encounters a significant obstacle known as the spring predictability barrier (SPB): predictions made later than May demonstrate good skill in predicting summer sea ice, while predictions made on or earlier than May exhibit considerably lower skill. This study develops a transformer-based deep-learning model, SICNet_season (V1.0), to predict the Arctic sea ice concentration on a seasonal scale. Including spring sea ice thickness (SIT) data in the model significantly improves the prediction skill at the SPB point. A 20-year (2000-2019) testing demonstrates that the detrended anomaly correlation coefficient (ACC) of Sep. sea ice extent (sea ice concentration > 15%) predicted by our model at May/Apr. is improved by 7.7%/10.61% over the ACC predicted by the state-of-the-art dynamic model from the European Centre for Medium-Range Weather Forecasts (ECMWF). Compared with the anomaly persistence benchmark, the mentioned improvement is 41.02%/36.33%. Our deep learning model significantly reduces prediction errors of Sep.'s sea ice concentration on seasonal scales compared to ECMWF and Persistence. The spring SIT data is key in optimizing the SPB, contributing to a more than 20% ACC enhancement in Sep.'s SIE at four to five months lead predictions. Our model achieves good generalization in predicting the Sep. SIE of 2020-2023.

## 1 Introduction

Arctic sea ice plays a significant role in the global climate because it modulates the thermal and dynamic exchanges between the ocean and the atmosphere(Ding et al., 2017; Kapsch et al., 2013; Liu et al., 2021a; Olonscheck et al., 2019). In recent decades, global warming has resulted in a dramatic retreat in Arctic sea ice during the summer and fall (Cao et al., 2017; Shu et al., 2022). This decline triggers a system-positive feedback mechanism that causes the Arctic's surface air temperature to increase 2-4 times faster than the global mean state, known as the Arctic amplification (AA) (England et al., 2021; Pithan and Mauritsen, 2014; Screen et al., 2013; Screen and Simmonds, 2010). AA accelerates sea ice decline, strengthening positive feedback (Jenkins and Dai, 2021; Kumar et al., 2010). If the situation is unchanged, climate models project that the



Arctic will become ice-free during summer by the 2050s (Andersson et al., 2021). The dramatic Arctic sea ice loss has led to consequences for global climate (Francis and Vavrus, 2012) and commercial activities (Min et al., 2022). For example, it weakens the stratospheric polar vortex in the winter, increasing extreme cold events in the Northern Hemisphere (Blackport et al., 2019; Cohen et al., 2014). Furthermore, the lower sea ice area during summer extends the navigability of the Arctic Passage to seasonal scales (Cao et al., 2022).

Sea ice predictions are helpful for better understanding global climate change and support human activities in the Arctic (Lindsay et al., 2008; Merryfield et al., 2013). Therefore, sea ice prediction, commonly represented by parameters such as sea ice concentration (SIC) or sea ice extent (SIE, defined as the sum of grid cell area where SIC>15%), has always attracted substantial efforts (Guemas et al., 2016; Stroeve and Notz, 2015). Various prediction systems are proposed, such as numerical  (Chevallier et al., 2013; Liang et al., 2020; Mu et al., 2020; Wang et al., 2013; Yang et al., 2019; Zhang et al., 2008, 2022), statistical (Gregory et al., 2020; Wang et al., 2016, 2022; Yuan et al., 2016), and deep learning models (Jun

Kim et al., 2020; Ren et al., 2022; Ren and Li, 2023). However, accurate sea ice prediction for Arctic summer remains challenging, particularly at seasonal or even longer scales (Zampieri et al., 2018). One of the biggest challenges is the spring predictability barrier (SPB): predictions for summer sea ice made before or on May show significantly lower skill than predictions made after May (Bonan et al., 2019; Bushuk et al., 2020; Day et al., 2014; Zeng et al., 2023). The Sea Ice

Prediction Network (SIPN) for the Sep.' SIE in the Arctic showed that individual dynamical and statistical predictions could not beat the anomaly persistence benchmark when predictions are made from early June to early August, with a root mean squared error (RMSE) of 0.5–0.7 million $km^2$  (Blanchard-Wrigglesworth et al., 2015, 2023). Studies show that SPB is evident in nearly all the fully coupled global climate models (GCMs) in Phase 5 of the Coupled Model Intercomparison Project (CMIP5), a crucial initiative providing climate projections to support essential climate research worldwide

(Blanchard-Wrigglesworth et al., 2011; Tietsche et al., 2014). So, optimizing the SPB is an urgent task for accurate summer sea ice predictions.

Experiments based on ensemble simulations reveal that the predictability of summer SIE is limited before spring due to the ice motion and growth in winter. However, the predictability increases rapidly after the melting processes in the spring (Bushuk et al., 2020). The satellite observations show that the spring sea ice thickness (SIT) correlates more with the summer SIE than

the spring SIE (Landy et al., 2022). These findings indicate that the spring SIT may be a key factor in optimizing the SPB (Bushuk et al., 2020, n.d.). Recently, researchers have assimilated the CryoSat-2 observed SIT data, the first summer SIT observations, into the Geophysical Fluid Dynamics Laboratory (GFDL) ocean–sea ice model and found that the prediction skill of Sep.'s SIC is improved significantly when the model is initialized with SIT anomaly in Jul. and Aug. (Zhang et al., 2023). This study further proves that the summer SIT data contributes to Sep's sea ice prediction. However, as the SPB flag is May for

most studies, whether the SPB could be optimized or even overcome by including SIT data remains largely unknown.

Currently, numerical models account for the mainstream sea ice prediction, but they are inflexible and have been limited by the SPB (Msadek et al., 2014; Sigmond et al., 2013). Statistical models are good at long-term prediction but cannot model complex nonlinear relationships and face SPB challenges. Deep learning models are more flexible than numerical models and



more powerful than traditional statistical ones, and they have been successfully used in Earth prediction problems (Li et al., 2021; Reichstein et al., 2019). Researchers have successfully developed deep-learning models to predict polar sea ice state from synoptic to sub-seasonal scales (Andersson et al., 2021; Dong et al., 2024; Li et al., 2024; Mu et al., 2023; Palerme et al., 2024; Ren et al., 2022; Ren and Li, 2023; Song et al., 2024; Wu et al., 2022; Yang et al., n.d.; Zhu et al., 2023), bringing the new potential to solve the SPB problem to improve the seasonal prediction skill from a data-driven perspective.

This work develops a seasonal sea ice prediction model named SICNet$_{season}$ (V1.0) to optimize the SPB. SICNet$_{season}$ is a transformer-based deep learning model with a physically constrained loss function based on SIC morphology. It takes historical SIC and SIT data as predictors and predicts the SIC of the following six months. The SIC data is the satellite-observed data from the National Snow and Ice Data Center (NSIDC) (Cavalieri et al., 1996). The SICNet$_{season}$ model is trained on data from 1979-2019 and tested with data from 2000-2019 by a leave-one-year-out strategy. Data from the recent four years, 2020-2023, is employed to verify the model's generalization. Experiments demonstrate that our model significantly optimizes the SPB with a higher detrend correlation coefficient (ACC) compared with anomaly persistence (Persistence) and the state-of-the-art dynamic predictions from the European Centre for Medium-Range Weather Forecasts (ECMWF) (Johnson et al., 2019). Our model significantly reduces the errors of Sep. SIC/SIE in four to five months lead predictions than the two compared models. The spring SIT data plays a key role in optimizing the SPB. Our model generalized well in predicting the Sep. SIE of 2020-2023. Finally, a comparison between our and the published deep learning model IceNet is discussed.

## 2 Data

### 1.1 Sea ice concentration data

The SIC data of 1979-2023 are experiment data. The SIC is the satellite-observed data obtained from the NSIDC. It is a daily observation derived from the Nimbus-7 Scanning Multichannel Microwave Radiometer (SMMR) and the Defense Meteorological Satellite Program (DMSP) Special Sensor Microwave Imager (SSM/I and SSMIS) (Cavalieri et al., 1996). The projection of the SIC data is the north-polar stereographic with a 25 km spatial resolution.

### 1.2 Sea ice thickness data

The SIT data is the reanalysis SIT from the Pan-Arctic Ice Ocean Modeling and Assimilation System (PIOMAS). PIOMAS is a numerical model with sea ice and ocean components, and it assimilates SIC and sea surface temperature (Zhang and Rothrock, 2003). PIOMAS SIT agrees well with in situ, airborne, and satellite measurements (Schweiger et al., 2011). It is daily data with an 18 km spatial resolution and is widely adopted by Arctic studies (Collow et al., 2015; Kwok et al., 2020; Nakanowatari et al., 2022). The SIC and PIOMAS SIT data are converted to a Northern Polar Stereographic Grid with 80 km resolution. The temporal resolution is one month.





## 3 Method

### 3.1 Framework of SICNet_season

The SICNet_season model is derived from a transformed-based U-Net deep learning model, SwinUNet (Cao Huand Wang, 2023). It accepts a three-dimensional sea ice data sequence and predicts a three-dimensional SIC sequence of the future, Fig. 1. The predicted target is the SIC of the next six months. For example, if we make predictions in May, the six months' predictions will cover the full summer and fall, from Jun. to Nov.

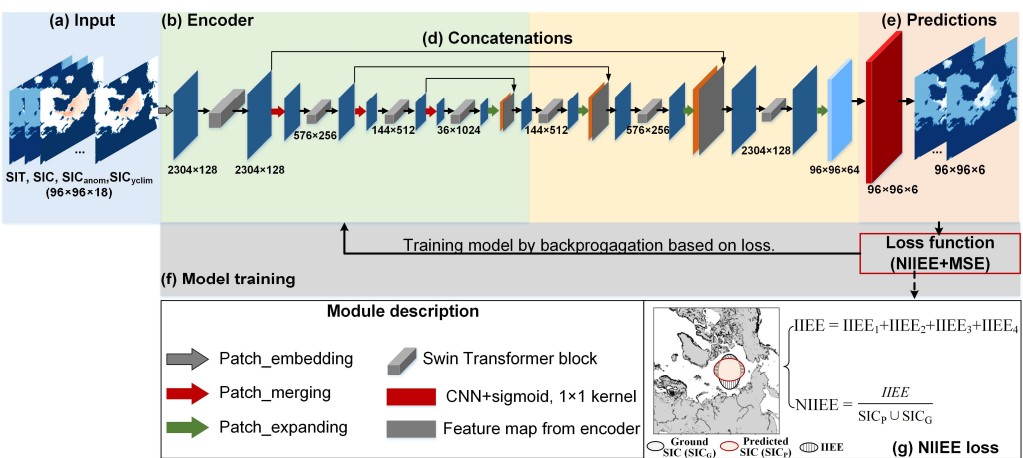

**Figure 1. Framework of model SICNet_season. (a) Input consists of SIT of the last three months, SIC of the last six months, SIC anomaly of the last three months, and SIC climatology of six target months, 96×96×18. (b) The encoder comprises four swin-transformer blocks and three patch-merging operators. (c) The decoder contains three swin-transformer blocks and four patch-expanding operators. (d) Concatenations connect the feature maps from the encoder and the decoder module. (e) A CNN layer with sigmoid activation transforms the feature map to the predicted SIC of six-month leads. (f) Model training procedure. The loss**
**function is combined with the normalized integrated ice-edge error (NIIEE) and the mean square error (MSE).**

The input for SICNet_season is a 96×96×18 SIC and SIT sequence, composed of SIT of the last three months, SIC of the last six months, SIC anomaly of the last three months, and SIC climatology of the six target months (Fig. 1a). The lengths of the input factors are determined by lots of experiments. The input is fed into the encoder to capture spatiotemporal correlations
among SIC/SIT data sequences at different levels to form multi-scale correlation maps. The encoder comprises four swin-transformer (Liu et al., 2021b) blocks and three patch merging operators (Fig. 1b). A swin-transformer block is a transformer unit integrated with shifted windows (Liu et al., 2021b). A transformer operator captures the global dependencies by attention mechanism. The shifted windows help the transformer operator capture local dependencies like the convolution operator. Therefore, local and global spatiotemporal dependencies among sea ice sequences can be captured. The patch
merging operator downscales the captured feature maps like the pooling layer in CNN models. The decoder upscales the feature maps through the patch expanding operator and swin-transformer blocks (Fig. 1c). The extracted correlation maps of





the encoder and decoder are stacked to form fused spatiotemporal maps (Fig. 1d). A CNN layer transforms the decoded feature maps to the same shape as the target SIC sequence. Here, it is a 96×96×6 array (Fig. 1e). As the range of SIC is 0-1, we employ the sigmoid function to activate the last feature map to transform the predicted values to 0-1.

During the training procedure, the loss is calculated between the predicted values and the ground values. Then, the model's parameters are trained by minimizing the loss value literately. We will explain the loss function in the following section.

### 3.2 Integrated ice-edge-constrained loss function

For a deep-learning model, the loss function is crucial during the training procedure as it guides the optimization of the
model's parameters. Here, the loss is the difference between the predicted values and the ground ones from NSIDC. Generally, the mean square error (MSE) is a basic loss function for prediction tasks. The MSE measures the mean state for all predicted values and cannot reflect the spatial differences between 2-dimensional SIC patterns. To address the issue, we proposed a normalized integrated ice-edge error (NIIEE) loss function that considers the spatial distribution of SIC to constrain the model's optimization (Fig. 1g).

The NIIEE loss is based on the integrated ice-edge error (IIEE), a common metric for sea ice predictions. The IIEE represents the error regions that are both overestimated and underestimated by the prediction model (Goessling et al., 2016). It measures the spatial similarity between two 2-dimension SIC patterns. Originally, the IIEE binaries the SIC by 15% to describe the SIE. For the SIC prediction here, we do not perform binarization. Let $P_{SIC}/G_{SIC}$ represent the predicted/ground SIC; the IIEE is calculated by Eq. (1). We normalize IIEE to the range of 0-1 to form the NIIEE loss by Eq. (2). If the NIIEE
loss is 0, the predicted SIC and the ground SIC will match in spatial and numerical. The fundamental MSE loss has been demonstrated to be effective for prediction tasks. If the number of all predicted values is $N$, the MSE is calculated by Eq. (3). We combine the NIIEE with MSE as the loss function of the SICNet$_{season}$. A constant scale factor, here 0.01, is multiplied by NIIEE to balance its range with that of MSE, Eq. (4).

$$IIEE = (P_{SIC} \cup G_{SIC}) - (P_{SIC} \cap G_{SIC}) \tag{1}$$

$$NIIEE = \frac{IIEE}{P_{SIC} \cup G_{SIC}} = 1 - \frac{P_{SIC} \cap G_{SIC}}{P_{SIC} \cup G_{SIC}} \tag{2}$$

$$MSE = \frac{\Sigma(P_{SIC} - G_{SIC})^2}{N} \tag{3}$$

$$Loss = 0.01 \times NIIEE + MSE \tag{4}$$



## 4 Experiments

### 4.1 Model training

The model is trained on a computer station with an NVIDIA Tesla V100 32-GB card. The training and test samples are constructed by step-by-step sliding. The testing period is 2000-2019. The leave-one-year-out strategy is adopted to train/evaluate our SICNet$_{season}$ model. For example, if the testing year is 2000, the training set is from 1979-1999 and 2001-2019. The validation set is split 20% from the training set. We set the batch size as eight, and the initial learning rate as 150 0.0001. We employ the early stopping strategy to break the training procedure when the validation loss does not decrease. The model is trained three times to eliminate random errors. The testing set is run on three trained models, and the mean values are adopted as the final predictions. Data from the recent four years, 2020-2023, is employed to verify the model's generalization. Data from these four years did not participate in the training stage. They are fed into the trained models obtained by the leave-one-year-out strategy to get the predictions. The predictions are the mean values of the 20 trained 155 models.

### 4.2 Evaluation metrics

The mean absolute error (MAE), Binary Accuracy (BACC), and detrend ACC are evaluation metrics. The MAE is for SIC, and the other two metrics are for SIE. To accurately calculate the metrics, we use the maximum observed monthly SIE since 1979 to mask the predictions. Assuming the predicted/truth value of the $i_{th}$ grid is $p_i/g_i$, the number of validation grids is $N$. 160 The MAE values are calculated by (5). The BACC of time $t$ is obtained by using one to subtract the ratio of IIEE to the area of the activated grid cell region (the maximum observed SIE during 1979-2019) of $t$ by equation (6). The detrend ACC of SIE is the anomaly correlation coefficient of two detrend SIE series. Each SIE series has 20 elements, years of 2000-2019.

$$MAE = \frac{\sum_1^N |p_i - g_i|}{N} \tag{5}$$

$$BACC = (1 - \frac{IIEE}{\text{area of the activated grid cell region}}) \times 100\% \tag{6}$$

### 4.3 Model skill in seasonal predictions


We compare the SICNet$_{season}$ with the Persistence and the ECMWF dynamical model SEAS5 to validate our model's ability to optimize the SPB. The Persistence model estimates the target SIC values by adding the current anomaly to the climate mean state at the target time, widely adopted as a benchmark for sea ice prediction (Blanchard-Wrigglesworth et al., 2023). The SEAS5 is a new seasonal forecast system showing excellent sea ice prediction skills from the ECMWF (Johnson et al., 170 2019). A BACC value of 100% indicates that the predicted SIE matches the observed SIE 100% in spatial. The metrics are calculated on 20 testing years, 2000-2019, in a leave-one-year-out training/testing strategy. As the SPB occurred to the target month in summer, we focus on the four summer months, Jun. to Sep.



Fig. 2 shows the detrended ACC and BACC of target months, Jun.-Sep., on six lead months' predictions. As shown in Figs. 2(a) and (b), the predictions of Persistence and ECMWF show an apparent SPB: the detrended ACC drops sharply when the predictions are made earlier than May, with a maximum ACC gap between two adjacent lead months marked by black lines. Taking the Sep. for example, the detrend ACC is 56.39% for Persistence when the prediction is made in Jun. (three months lead). Then, it decreases to 26.59% in May's prediction (four months lead), Fig. 2(a). For ECMWF, the ACC of Jun.'s prediction is 83.94%, then drops to 59.91% at May's prediction, forming a 24.03% ACC gap, Fig. 2(b). Although the SICNet$_{season}$'s prediction also shows an SPB feature, the black line in Fig.2 (c), the ACC at May's prediction is improved to 67.61%, and the ACC gap is reduced to 15.6%. Further, the ACC difference is calculated between SICNet$_{season}$ and Persistence/ECMWF. Compared with Persistence, SICNet$_{season}$ improves the ACC in most predictions, Fig. 2d. The ACC improvements along the SPB flag are more than 30% on average (the lead months right to the black line in Fig. 2d). Compared with the ECMWF, SICNet$_{season}$ also improves the prediction skill of the SPB. When the target months are Jun. and Jul., SICNet$_{season}$ shows a much higher prediction skill than ECMWF in four to six months lead predictions, Fig. 2e. For the target month Sep., the SICNet$_{season}$ improves the ACC by 7.70%/10.61% than ECMWF when prediction is made in May/Apr. (four/five months lead in Fig. 2e). For the target month Aug./Sep., the SICNet$_{season}$ shows lower ACCs than the ECMWF when prediction is made on or before Mar. (five/six months lead for Aug./Sep.). However, for the predictions made adjacent to the SPB flag line, the SICNet$_{season}$ achieves larger ACCs than the ECMWF (values right to the black line in Fig. 2e). Therefore, the SICNet$_{season}$ optimizes the SPBs significantly compared to the well-known numerical model.

The BACC of ECMWF also shows a similar SPB characteristic to the ACC. A sharp BACC drop occurred when the prediction was made on and before May, the black line in Fig. 2(g). The maximum BACC gaps of Persistence and SICNet$_{season}$ occurred in the second lead month. However, the maximum BACC gap of SICNet$_{season}$ is about 2%, much lower than the 10% gap of Persistence and ECMWF. Compared with the Persistence and ECMWF, the SICNet$_{season}$ improves the BACC by more than 10% in predicting SIE of Aug. and Sep. three-six months' lead, Figs. 2(i) and (j).

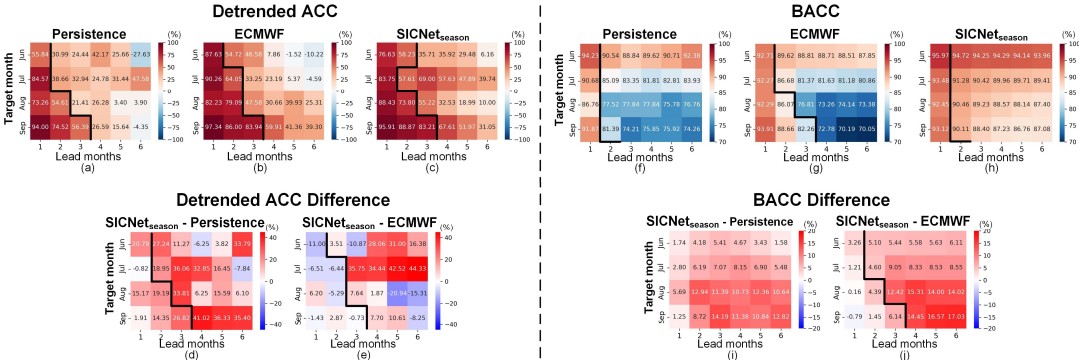

**Figure 2. Detrend ACC of SIE, BACC of SIE, and their differences of Persistence, ECMWF, and SICNet$_{season}$ from Jun. to Sep., averaged by 2000-2019. (a)-(c) Detrend ACC of three models. Two detrend SIE series (predicted and observed) calculate each**





value. (d)-(e) Detrend ACC differences between SICNet$_{season}$ and Persistence/ECMWF. (f)-(h) BACC of three models. Each BACC is a mean value during 20 testing years. (i)-(j) BACC differences of SICNet$_{season}$ and Persistence/ECMWF. The black line indicates the SPB: a maximum decrease between two adjacent lead months.

**4.4 Performance in predicting SIC of Sep.**

As September's sea ice draws wider attention than other months, we calculate the MAE of SIC of Sep. predicted by three models. Fig. 3 shows the spatial MAE of Persistence, ECMWF, and SICNet$_{season}$ on six lead months. The MAE in the three models is not much different for the first two lead months. When the lead month is one, the MAE of ECMWF is slightly better than that of Persistence and SICNet$_{season}$, indicating that the dynamic model performs well in monthly predicting. However, when the lead month is longer than three, the ECMWF's MAEs are much more than 45% in the Pacific sector, mainly containing the Beaufort Sea, the Chukchi Sea, the East Siberian Sea, and the Laptev Sea, Figs. 3(j)-(l). The SICNet$_{season}$ reduces the MAEs to 20-30% for most regions in the Pacific Arctic, Figs. 3(q)-(r). Compared with Persistence, SICNet$_{season}$ also reduces MAEs by 5-10% in the mentioned four local seas, Figs. 3(d)-(f). Therefore, the SICNet$_{season}$ significantly reduces the SIC errors of Sep. in seasonal scale predictions (three to six months lead).

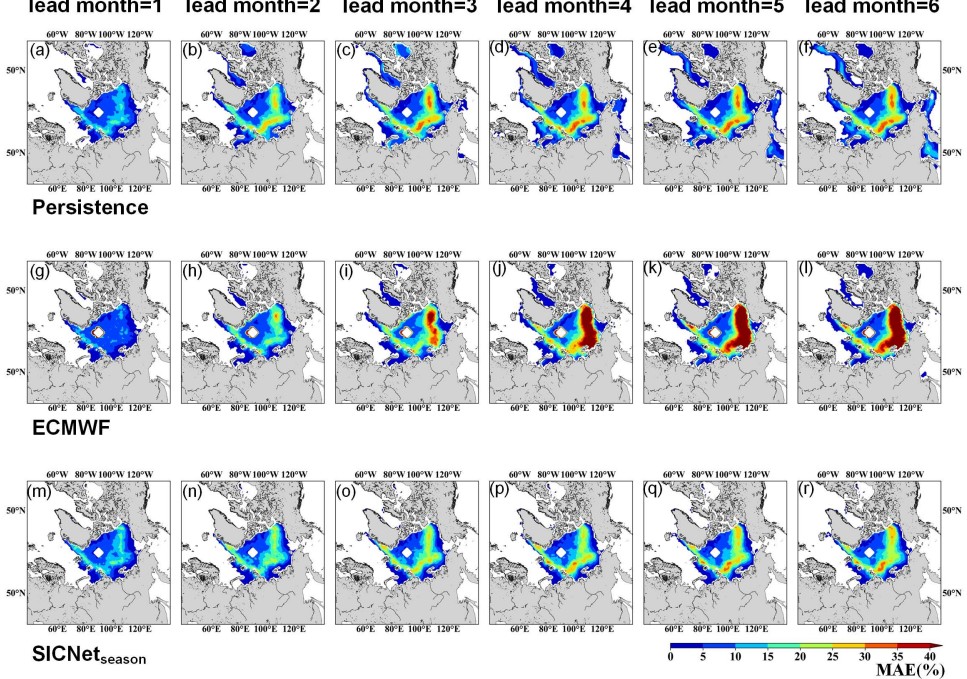

Figure 3. The MAEs of September's predictions from three compared models: each value is averaged by 20 testing years. (a)-(g) MAEs of Persistence. (h)-(m) MAEs of ECMWF. (n)-(s) MAEs of SICNet$_{season}$.




**4.5 SIT's contributions to seasonal predictions**

We further conduct a comparison experiment to validate the role of SIT data in seasonal predictions based on SICNet$_{season}$. The model without SIT as an input is named SICNet$_{season\_nosit}$. The other settings for SICNet$_{season\_nosit}$ are the same as those for SICNet$_{season}$. The detrended ACC and BACC are shown in Fig. 4.

Without the SIT data as input, the model's prediction skill drops apparently in three to six months lead predictions, Fig. 4(a). For the target month Sep., the detrend ACC is 76.41% when the prediction is made in Jun. (three months' lead). Then,

the ACC drops to 26.43% at May's prediction (four months' lead). By including SIT data as input, the ACC of May's prediction is improved by 41.18% in model SICNet$_{season}$, Fig. 4(c). For the target month Aug., the ACC improvement at May's prediction (three months' lead) by including SIT data is 42.44%. Therefore, the SIT data is important to improve the model's prediction skill on SPB.

For target months Aug. and Sep., the BACCs of SICNet$_{season\_nosit}$ show an apparent drop in three to six months lead

predictions. By including SIT data as the model's input, the BACC improvement is 0.95%/2.02% for the target month Aug./Sep. at May's predictions, Fig. 4(f). Then, we calculate the MAE of the target month Sep., Fig. 5. The MAEs of the first two lead months are similar for the two models. When the lead month is larger than three, the MAEs of SICNet$_{season\_nosit}$ in the Beaufort Sea, the East Siberian Sea, and the Laptev Sea are 30-45%, red cycles in Figs. 5(d)-(f). By including SIT data, the MAEs in the mentioned three regions are reduced to 20-35% by SICNet$_{season}$, red cycles in Figs. 5(j)-(l). Therefore,

including SIT data reduces the errors of Sep.'s SIC by more than 10% in seasonal scale predictions.

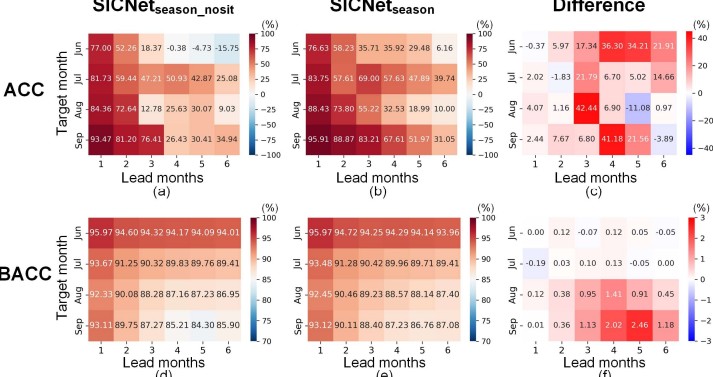

**Figure 4. Detrended ACC of SICNet$_{season\_nosit}$ (a) and SICNet$_{season}$ (b). (c) ACC difference obtained by SICNet$_{season}$ minus SICNet$_{season\_nosit}$. BACC of SICNet$_{season\_nosit}$ (d) and SICNet$_{season}$ (e). (f) BACC difference like (c).**



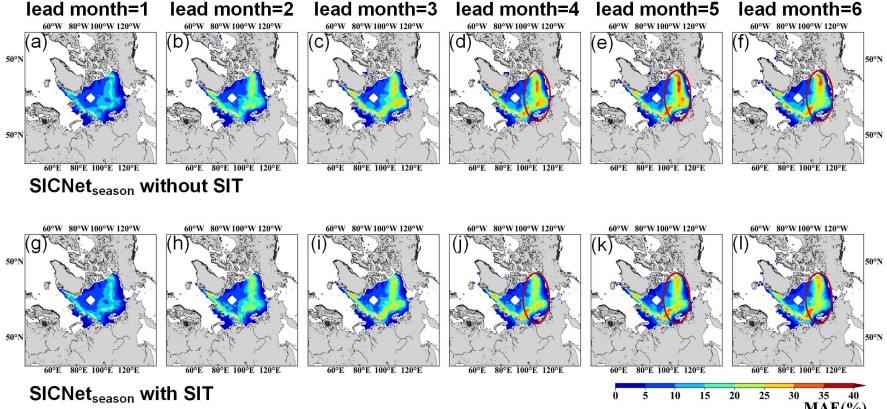


**Figure 5. The MAEs of Sep.'s SIC predicted by SICNet$_{season}$ and SICNet$_{season\_nosit}$. Each value is averaged by 20 testing years. (a)-(g) MAE of SICNet$_{season\_nosit}$. (h)-(m) MAE of SICNet$_{season}$. The red cycles marked the regions where the MAE is reduced typically by including SIT data.**

**4.6 Generalization in predicting the SIEs of 2020-2023**

To verify our model's generalization in predicting the SIEs of recent years, we employed the twenty trained models to predict the SIE of the recent four years, 2020-2023. The twenty models are the trained models for 2000-2019 mentioned in the earlier sections. The data from 2020-2023 is "blind" for the models. The mean values of the twenty models' predictions are the final predictions. As the temporal span of four years is too short for calculating ACC, we use the BACC as the metric (Fig. 6). Compared with Persistence and ECMWF, the SICNet$_{season}$ achieves higher BACCs in predicting SIEs of Aug. and

Sep. For the target month Sep., the BACC of SICNet$_{seaon}$ is 10% higher than those of the other two models in three to six months lead predictions.

We draw the observed and predicted SIEs of 2020-2023 in Fig. 7. The ECMWF model achieves good accuracy when the lead month is one, with the largest BACC values in 2020-2022, Fig. 7(a)/(g)/(m). This result proves that the dynamic model is good at short-term predictions. However, for the lead month longer than two, our data-driven model shows obvious

advantages that the ECMWF and Persistence. For predictions made on or before May, lead months of four to six, the BACCs of SICNet$_{season}$ are much higher than those of Persistence and ECMWF, Fig. 7. The SIE in 2020/2023 is the second/sixth lowest record in the Arctic since 1979. At May's prediction, our model achieved a BACC of 82.25%/82.08%, about 12%/10% higher than Persistence and ECMWF, Fig. 7(d)/(v). Therefore, the SICNet$_{season}$ model achieves good generalization ability in predicting the SIEs of 2020-2023.



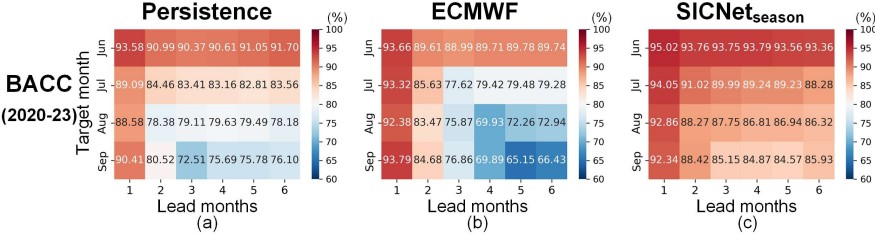


**Figure 6. BACC of 2020-2023. (a) Persistence, (b) ECMWF, and (c) SICNet_season. Each value is a mean value of the four testing years. The horizontal axis represents the six lead months, and the vertical axis represents the target months, Jun. to Sep.**

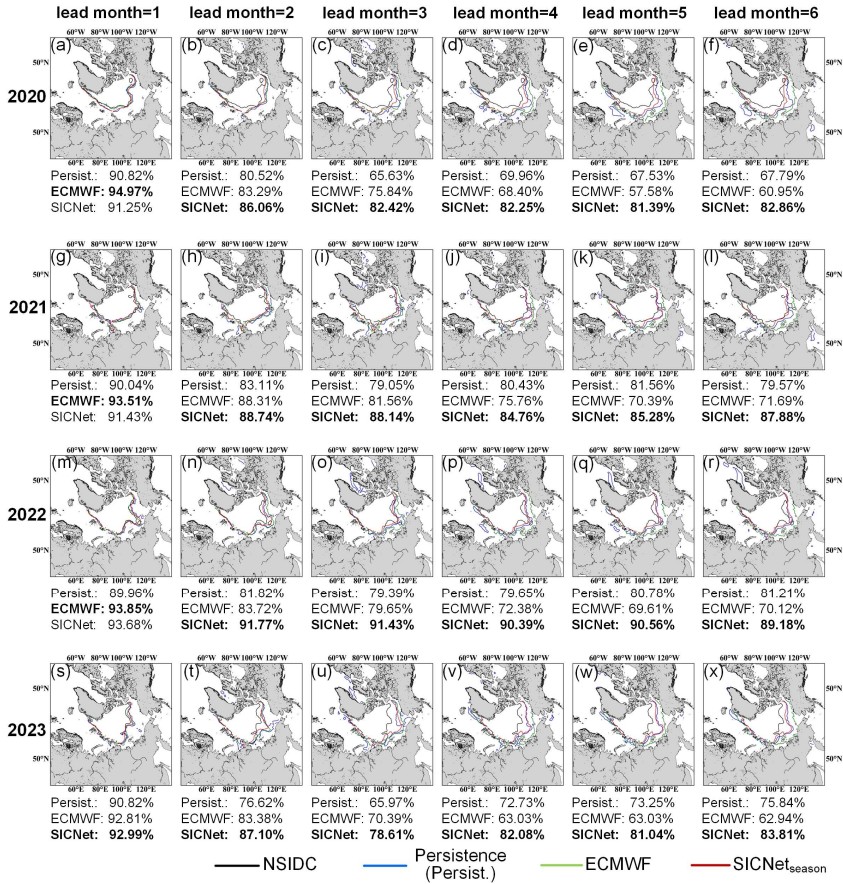



**Figure 7. Predicted Sep. SIEs and their BACCs of 2020-2023 by Persistence, ECMWF, and SICNet_season. (a)-(f) 2020, (g)-(l) 2021, (g)-(l) 2022, and (s)-(x) 2023.**

### 4.7 Comparison with the representative deep learning model

We compare the SICNet_season against the representative deep learning sea ice prediction model, IceNet. The IceNet is a seasonal sea ice prediction model and achieves state-of-the-art performance in SIE prediction (Andersson et al., 2021). It is a

CNN-based classification model, and it outputs the probability of three classes: open water (SIC≤15%), marginal ice (15% < SIC < 80%), and full ice (SIC≥80%). Differently, our SICNet_season outputs the 0-100% range SIC values. The IceNet's inputs consist of 50 monthly mean variables, including SIC, 11 climate variables, statistical SIC forecasts, and metadata.

To make a fair comparison, we set the inputs (including SIT data) of the IceNet to the same ones as SICNet_season. The loss function is also set as the NIIEE+MSE. We transform the original IceNet's output to the continuous values of 0-100%. We

also change the number of CNN filters to make the number of training parameters in IceNet equal to that in SICNet_season, about 140 million. The training and testing settings of IceNet are the same as those of SICNet_season.

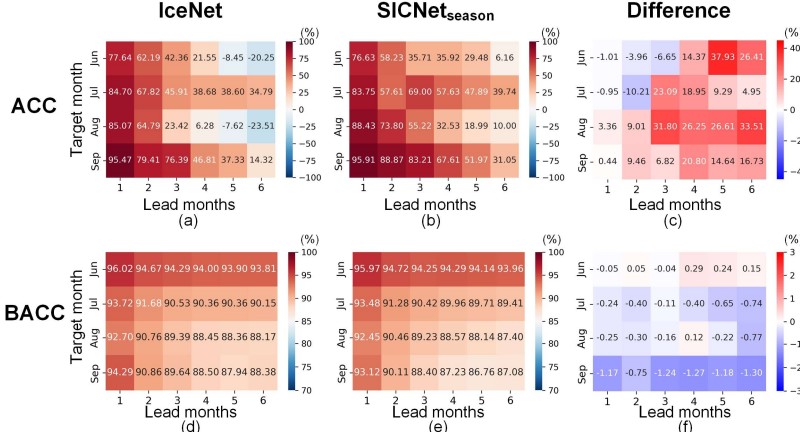

**Figure 8. Detrended ACC of IceNet (a) and SICNet_season (b). (c) ACC difference obtained by SICNet_season minus IceNet. BACC of IceNet (d) and SICNet_season (e). (f) BACC difference like (c).**




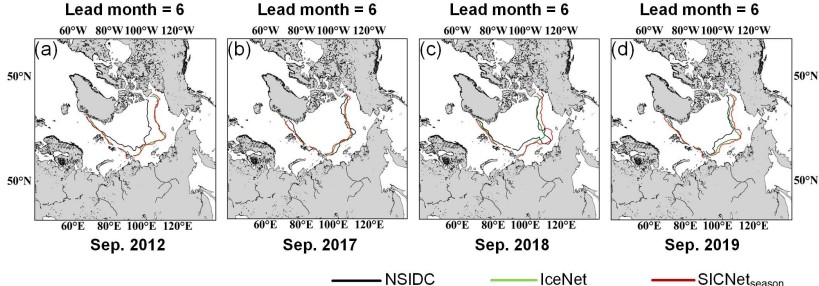

**Figure 9. The predicted Sep. SIEs of IceNet and SICNet$_{season}$ in six months' lead: (a) 2012, (b) 2017, (c) 2018, and (d) 2019. The black edge is the NSIDC observed SIE, the green one is IceNet's SIE, and the red one is SICNet$_{season}$'s SIE.**

Fig. 8 shows the detrend ACC, BACC, and the differences between the two models. Compared with the IceNet model, our SICNet$_{season}$ model significantly improves the ACC at most predictions, Fig. 8(c). For the target month, Aug./Sep., the SPB feature is obvious in the IceNet: the maximum ACC gap is about 40%/30% at predictions made in May and Jun., Fig. 8(a). Our SICNet$_{season}$ model optimizes the ACC gap with an improvement of 31.8%/20.8% at May's predictions, Fig. 8(c). The ACC improvements are also larger than 15% for predictions made before May. Therefore, compared with the state-of-the-art deep learning model IceNet, our model achieves more skillful seasonal predictions by optimizing the SPB.

Unlike the ACC values, the BACC values of IceNet are larger than those of SICNet$_{season}$ on most predictions, Fig. 8(f). This result implies that IceNet is more dependent on SIE trends than SICNet$_{season}$. This difference can be attributed to the distinct fundamental units employed by the two models. The IceNet is a CNN-based model, and the weight-sharing mechanism of convolutional kernels forces the model to capture the most "common" local dependencies in spatial. Though representative, these "common" local dependencies tend to yield smoother model outputs. The SICNet$_{season}$ is a transformer-based model. The attention mechanism of the transformer can capture global dependencies without weight-sharing. As a result, "personalized" global dependencies are extracted and the output is not smoothed like the output of a CNN-based model. The "common" local dependencies have more apparent trend features than the "personalized" global dependencies. Fig. 9 shows the Sep.'s SIEs predicted by IceNet and SICNet$_{season}$ in the sixth-month lead. The SIEs of IceNet are smoother than those of SICNet$_{season}$. For 2012 and 2017, the SIEs' locations of the two models are very similar. For the other two years, the SIEs of IceNet match the observed SIEs better than those of SICNet$_{season}$. However, the SIEs of IceNet are over-smoothed and fail to characterize some abnormal characteristics, such as the SIE in Sep. 2018, Fig. 9(c).

Therefore, our transformer-based SICNet$_{season}$ is more skillful than the representative CNN-based model IceNet in optimizing the SPB. The SICNet$_{season}$ exhibits a lower dependency on SIE trends and lower smooth results than the CNN-based model.

**5 Conclusion**

This study develops a deep-learning model, SICNet$_{season}$, to predict the Arctic SIC on a seasonal scale. The model is derived





from a SwinUNet architecture. It inputs the historical SIC, SIT, and SIC climatology of target moths and predicts the SIC of the next six months. A spatially constrained loss function NIIEE is employed to train the model considering sea ice distribution. We employ a 20-year (2000-2019) testing set to validate the model's performance. The summer season, Jun. to Sep., is chosen as the target period. The detrend ACC, BACC, and MAE are metrics. Comparison experiments with Persistence and seasonal predictions of ECMWF are made to validate our model's performance. In particular, an ablation experiment is carried out to investigate the role of SIT data in optimizing the SPB. A generalization experiment with recent four years' data, 2020-2023, is carried out. The seasonal predictions of Sep. SIEs are analyzed. Finally, we discuss the advantages and disadvantages between our model and the typical CNN-based model IceNet. Given the mentioned efforts, our study draws the following conclusions.

First, our deep learning model, SICNet$_{season}$, is skillful in predicting the Arctic sea ice seasonally. Compared with the dynamic model ECMWF, SICNet$_{season}$ optimizes the SPB significantly. The detrended ACC of Sep. SIE predicted by SICNet$_{season}$ in May/Apr. is improved by 7.7%/10.61% over the ACC predicted by the ECMWF. Compared with the anomaly persistence benchmark, the mentioned improvement is 41.02%/36.33%. Our deep learning model significantly reduces prediction errors of Sep.'s SIC on seasonal scales compared to ECMWF and Persistence, a 20-30% reduction measured by MAE.

Second, the spring SIT data is key in optimizing the SPB, contributing to a more than 20% ACC enhancement in Sep.'s SIE at four to five months lead predictions. By including SIT data, the MAEs in the Beaufort Sea, the East Siberian Sea, and the Laptev Sea are reduced by more than 10% compared with those without SIT data.

Third, our model achieves good generalization in predicting the Sep. SIEs of 2020-2023. When predicting the Sep.'s SIE in 2020/2023 (second/sixth lowest record) in May, SICNet$_{season}$ achieved a BACC of 82.25%/82.08%, about 12%/10% higher than Persistence and ECMWF.

Fourth, our transformer-based SICNet$_{season}$ is more skillful than the CNN-based model IceNet in seasonal sea ice predictions. Our SICNet$_{season}$ model optimizes the ACC gap with an improvement of 31.8%/20.8% at May's predictions over the IceNet. The SICNet$_{season}$ exhibits a lower dependency on SIE trends and lower smooth results than the CNN-based model. This is due to the attention mechanism of the transformer operator extracting "personalized" global dependencies, while the CNN operator captures the most "common" local dependencies globally. The "common" local dependencies smooth the map and depend more on the trend than "personalized" ones.

**Code and data availability**

The satellite-observed sea ice concentration is obtained from the following site https://nsidc.org/data/NSIDC-0081/versions/1 (Cavalieri et al., 1996). The reanalysis of sea ice thickness data is open access (http://psc.apl.uw.edu/research/projects/arctic-sea-ice-volume-anomaly/data/model_grid) (Zhang and Rothrock, 2003). The seasonal predictions of SEAS5 are available from the ECMWF (https://cds.climate.copernicus.eu/cdsapp#!/dataset/seasonal-



monthly-single-levels?tab=form) (Johnson et al., 2019). The code of the developed deep learning model, SICNet_season, is
available at https://doi.org/10.5281/zenodo.14232859.

### Acknowledgment

This work was supported by the National Science Foundation of China (42206202), (U2006211), (42221005), Laoshan
Laboratory Innovation Project (LSKJ202202302), (LSKJ202204302), in part by the China-Portugal Xinghai "Belt and Road"
(2022YFE0204600).

### Author contribution

All authors designed the experiments and carried them out. Yibin Ren developed and evaluated the model. Xiaofeng Li
designed experiments and revised the manuscript. Yunhe Wang analyzed the experimental results.

### Competing interests

The authors declare no competing interests.

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
