# Peer review of "SICNetseason V1.0: a transformer-based deep learning model for seasonal Arctic sea ice prediction by incorporating sea ice thickness data"

_Geoscientific Model Development, 2024_

## Referee Comment (RC1)

General comments:

In summary, this research aims to develop a transformer-based U-net model to improve prediction skill of Arctic sea ice concentration at seasonal lead times. The primary novel contribution of this work is the use of a state-of-the-art deep learning architecture with a custom domain-specific loss function as well as including spring sea ice thickness as an input predictor to overcome the common spring predictability barrier (SPB) faced by previous numerical and DL-based models that decrease prediction skill of predictions made before May.

The authors choose to use a transformer based architecture to capture spatiotemporal patterns in the input data. The model inputs include multiple sea-ice related variables over 3-6 months and outputs the sea ice concentration of the next six months, specifically looking at predictions for June-Sept at 6 month lead times. The authors develop a novel loss function that combines both a standard DL loss function (mean squared error) and a domain-specific NIEE loss which accounts for spatial similarity in model predictions and ground truth. The model follows a standard DL training procedure and includes three years of unseen data to test their model performance. The authors evaluate their model using standard DL performance metrics along with Binary Accuracy (BACC) which accounts for accurate spatial distributions.

The authors succeed in improving prediction skill when compared to the benchmark Persistence model and dynamical ECMWF model. For Sept ice prediction specifically, the ECMWF model performs better at smaller lead times but the author's model shows marked performance improvement at a lead time of 3-6 months, thus overcoming the spring predictability barrier. The authors perform necessary ablation tests to investigate the role of sea ice thickness in their model. They show a model that includes sea ice thickness as input outperforms a model without sea ice thickness at seasonal lead times of 3-6 months. When testing their model on unseen data, the authors use an ensemble model approach by averaging the results of 20 separately trained models.The authors also test their model against a previous state of the art deep learning CNN based approach, IceNet. They show their model produces better ACC scores at longer lead times. Their model produced lower BACC scores but can capture more individualized/local or extreme characteristics in comparison to IceNet which tends to produce smoothed out results.

Overall, this manuscript provides a significant improvement to the modeling of sea ice concentration by providing a novel approach to increase prediction skill at seasonal lead times to overcome the SPB. The authors clearly state their motivations for the project and outline their novel contributions. The authors discuss previous modeling approaches in the field and clearly showcase their significant scientific results where their model outperforms previous approaches.The contributions of this paper is twofold, one in utilizing a novel transformer based approach that the authors state has not been previously used in this field and second in highlighting the impact of spring sea ice thickness on improving prediction accuracies, revealing potential scientific insights that need to be studied further. Below are a few suggestions to improve reproducibility and presentation quality.

Specific comments:

In line 52, the authors state that experiments show the reason for decrease in prediction skill before spring is due to ice motion and growth in the winter. It is unclear which experiments in the literature the authors are referring to. Including a citation here would help justify this statement.

In line 79, a citation for the IceNet model (Andersson et al 2021) is missing. Since working with the IceNet model is a significant portion of the manuscript, perhaps including more information about the IceNet model and how it differentiates from the author's transformer based model would provide more context to the reader in this section.

In lines 108-109, the authors state that multiple experiments were conducted to determine the length of the model input features. It is unclear what experiments or methods the authors tried. Was this determined by manual tuning using their cross-validation strategy or did the authors employ an automated grid-search type strategy. The authors can also include if domain knowledge or previous literature informed the choice of input lengths of the data.

In section 4.7, when comparing the transformer based model to the CNN-based IceNet model, the authors state they used identical training and testing settings to perform fair comparisons. It is unclear whether the authors used the same 20 trained ensemble model approach they had used for their transformer model for the IceNet model. If so, specifying whether they used an ensemble approach or singular-model approach for IceNet would clarify this for the reader. If the authors did not use a similar ensemble approach, the authors should justify this choice.

In line 269, it is unclear how the authors transformed the IceNet output to match the continuous scale (for e.g restructuring only the final output layer). Having this additional context would help with reproducibility of this experiment.

For Figures 2, 4, 6 and 8, it is slightly confusing to the reader at first how to interpret these results, specifically the Difference column. Perhaps including the caption that red signifies improvement in accuracy and blue signifies a decrease would aid in understanding especially because in Figures 3 and 5 the opposite color scheme is used (red = high error, blue - lower error).

Technical comments:

In line 20 of the abstract, 'SIE' is referenced without expanding the full word.

In line 24, there is a space missing before the in-text citation.

In Figure 2, the score percentages are hard to read and their text size could be increased.

---

## Author Comment (AC3)

**General comments:**

In summary, this research aims to develop a transformer-based U-net model to improve prediction skill of Arctic sea ice concentration at seasonal lead times. The primary novel contribution of this work is the use of a state-of-the-art deep learning architecture with a custom domain-specific loss function as well as including spring sea ice thickness as an input predictor to overcome the common spring predictability barrier (SPB) faced by previous numerical and DL-based models that decrease prediction skill of predictions made before May.

The authors choose to use a transformer based architecture to capture spatiotemporal patterns in the input data. The model inputs include multiple sea-ice related variables over 3-6 months and outputs the sea ice concentration of the next six months, specifically looking at predictions for June-Sept at 6 month lead times. The authors develop a novel loss function that combines both a standard DL loss function (mean squared error) and a domain-specific NIEE loss which accounts for spatial similarity in model predictions and ground truth. The model follows a standard DL training procedure and includes three years of unseen data to test their model performance. The authors evaluate their model using standard DL performance metrics along with Binary Accuracy (BACC) which accounts for accurate spatial distributions.

The authors succeed in improving prediction skill when compared to the benchmark Persistence model and dynamical ECMWF model. For Sept ice prediction specifically, the ECMWF model performs better at smaller lead times but the author's model shows marked performance improvement at a lead time of 3-6 months, thus overcoming the spring predictability barrier. The authors perform necessary ablation tests to investigate the role of sea ice thickness in their model. They show a model that includes sea ice thickness as input outperforms a model without sea ice thickness at seasonal lead times of 3-6 months. When testing their model on unseen data, the authors use an ensemble model approach by averaging the results of 20 separately trained models. The authors also test their model against a previous state of the art deep learning CNN based approach, IceNet. They show their model produces better ACC scores at longer lead times. Their model produced lower BACC scores but can capture more individualized/local or extreme characteristics in comparison to IceNet which tends to produce smoothed out results.

Overall, this manuscript provides a significant improvement to the modeling of sea ice concentration by providing a novel approach to increase prediction skill at seasonal lead times to overcome the SPB. The authors clearly state their motivations for the project and outline their novel contributions. The authors discuss previous modeling approaches in the field and clearly showcase their significant scientific results where their model outperforms previous approaches. The contributions of this paper is twofold, one in utilizing a novel transformer based approach that the authors state has not been previously used in this field and second in highlighting the impact of spring sea ice thickness on improving prediction accuracies, revealing potential scientific insights that need to be studied further. Below are a few suggestions to improve reproducibility and presentation quality.
**Response:** Thanks for the comment.

**Specific comments:**

**Comment 1**: In line 52, the authors state that experiments show the reason for decrease in prediction skill before spring is due to ice motion and growth in the winter. It is unclear which experiments in the literature the authors are referring to. Including a citation here would help justify this statement.

**Response:** Thanks for the comment. We cite the following reference in this sentence:

Bushuk, M., Winton, M., Bonan, D. B., Blanchard-Wrigglesworth, E., and Delworth, T. L.: A mechanism for the Arctic sea ice spring predictability barrier, Geophys Res Lett, 47, https://doi.org/10.1029/2020GL088335, 2020.

**Comment 2**: In line 79, a citation for the IceNet model (Andersson et al 2021) is missing. Since working with the IceNet model is a significant portion of the manuscript, perhaps including more information about the IceNet model and how it differentiates from the author's transformer based model would provide more context to the reader in this section.

**Response:** Thanks for the comment. We added the citation for IceNet in line 79. A brief introduction about IceNet has also been added. The revised sentences are as follows:

Finally, we compare our SICNet$_{season}$ model with the published deep learning model IceNet (Andersson et al., 2021). IceNet is a probability prediction model for Arctic SIE based on convolutional neural network (CNN) units and the U-Net architecture. It achieved state-of-the-art performance in predicting the SIE for six months (Andersson et al., 2021). Therefore, we chose IceNet as a comparison model.

Besides, more information about the IceNet model, such as the model's inputs and outputs, was presented in Section 4.7, lines 263-267:

The IceNet is a seasonal sea ice prediction model that performs state-of-the-art SIE prediction (Andersson et al., 2021). It is a CNN-based classification model, and it outputs the probability of three classes: open water (SIC≤15%), marginal ice (15% < SIC < 80%), and full ice (SIC≥80%). Differently, our SICNet$_{season}$ outputs the 0-100% range SIC values. The IceNet's inputs consist of 50 monthly mean variables, including SIC, 11 climate variables, statistical SIC forecasts, and metadata.

**Comment 3**: In lines 108-109, the authors state that multiple experiments were conducted to determine the length of the model input features. It is unclear what experiments or methods the authors tried. Was this determined by manual tuning using their cross-validation strategy or did the authors employ an automated grid-search type strategy. The authors can also include if domain knowledge or previous literature informed the choice of input lengths of the data.

**Response:** Thanks for the comment. We determine the length of input factors by combining domain knowledge and manual tuning experiments. The main domain knowledge we considered is the sea ice reemergence mechanism. The spring-fall reemergence occurs between pairs of months where the ice edge is in the same position, such as in May and December (Blanchard-Wrigglesworth et al., 2011; Day et al., 2014). The spring sea ice anomaly is positively correlated with fall sea ice anomalies, and there is also a weaker reemergence between fall sea ice anomalies and anomalies the following spring (Bushuk et al., 2015). Therefore, we set the initial input length of the SIC/SIT/SIC anomaly as six months. Then, we change the length manually to fine-tune the deep

learning model to find the best-matched length for each factor. The SIC climatology of the target months provides an essential mean state of the prediction SIC, so we input it into the model. It can also represent the month number signal that IceNet has considered. We explain more details about this issue in the revised manuscript, and the new revision is as follows:

The input for SICNet$_{season}$ is a 96×96×18 SIC and SIT sequence, composed of SIT of the last three months, SIC of the last six months, SIC anomaly of the last three months, and SIC climatology of the six target months (Fig. 1a). We determine the length of input factors by combining domain knowledge and manual tuning experiments. The primary domain knowledge we considered is the spring-fall reemergence mechanism. It occurs between pairs of months where the ice edge is in the same position, such as in May and December (Blanchard-Wrigglesworth et al., 2011; Day et al., 2014). The spring sea ice anomaly is positively correlated with fall sea ice anomalies, and there is also a weaker reemergence between fall sea ice anomalies and anomalies the following spring (Bushuk et al., 2015). Therefore, we set the initial input length of the SIC/SIT/SIC anomaly as six months. We change the input length manually (from six to one in step one) to fine-tune the deep learning model to find the best-matched length for each factor. The SIC climatology of the target months provides an essential mean state of the prediction SIC. It represents the monthly cycle signal that IceNet has considered.

[1] Blanchard-Wrigglesworth, E., Armour, K. C., Bitz, C. M., and Deweaver, E.: Persistence and inherent predictability of arctic sea ice in a GCM ensemble and observations, J Clim, 24, 231–250, https://doi.org/10.1175/2010JCLI3775.1, 2011.
[2] Day, J. J., Tietsche, S., and Hawkins, E.: Pan-arctic and regional sea ice predictability: Initialization month dependence, J Clim, 27, 4371–4390, https://doi.org/10.1175/JCLI-D-13-00614.1, 2014.
[3] Bushuk, M., Giannakis, D., and Majda, A. J.: Arctic Sea Ice Reemergence: The Role of Large-Scale Oceanic and Atmospheric Variability*, https://doi.org/10.1175/JCLI-D-14-00354.s1, 2015.

**Comment 4**: In section 4.7, when comparing the transformer based model to the CNN-based IceNet model, the authors state they used identical training and testing settings to perform fair comparisons. It is unclear whether the authors used the same 20 trained ensemble model approach they had used for their transformer model for the IceNet model. If so, specifying whether they used an ensemble approach or singular-model approach for IceNet would clarify this for the reader. If the authors did not use a similar ensemble approach, the authors should justify this choice.

**Response:** Thanks for the comment. We are sorry for the confusion. We did not use an ensemble approach for both SICNet$_{season}$ and IceNet. The training procedure is a leave-one-year-out strategy for the 20 testing years (2000-2019). For example, if the testing year is 2019, the training set is data from 1979-2018, and the testing data is 2019. Then, the testing year moves to 2018, and the corresponding training set is data from 1979-2017 and 2019. The model is trained 3 times for each training-testing pair to eliminate randomness, and the prediction for each testing year is the mean value of the three trained models. We explain the leave-one-year-out training procedure in Section 4.1. Further, we clarify the training strategy in the revision:

The training and testing settings of IceNet are the same as those of SICNet$_{season}$. The IceNet is trained using the same leave-one-year-out strategy as the SICNet$_{season}$. For example, if the testing year is

2019, the training set is data from 1979-2018, and the testing data is 2019. Then, the testing data moves to 2018; the remaining data (1979-2017, 2019) is the training set. For each training/testing pair, the model is trained three times to eliminate randomness, and the final prediction for testing data is the mean value of the three models.

**Comment 5**: In line 269, it is unclear how the authors transformed the IceNet output to match the continuous scale (for e.g restructuring only the final output layer). Having this additional context would help with reproducibility of this experiment.

**Response:** Thanks for the comment. We reconstruct IceNet's output layer by replacing the softmax with the sigmoid activation function. The sigmoid function outputs continuous values of 0-100%. We clarify this point in the revision:

We reconstruct IceNet's output layer by replacing the original softmax with the sigmoid activation function. The sigmoid function outputs continuous values of 0-100%.

**Comment 6**: For Figures 2, 4, 6 and 8, it is slightly confusing to the reader at first how to interpret these results, specifically the Difference column. Perhaps including the caption that red signifies improvement in accuracy and blue signifies a decrease would aid in understanding especially because in Figures 3 and 5 the opposite color scheme is used (red = high error, blue – lower error).

**Response:** Thanks for the comment. We have added the following statement to the captions of Figures 2, 4, 6, and 8: The red signifies a high/improvement in ACC/BACC, and the blue signifies a decrease.

---

## Author Comment (AC4)

**Summary**

The manuscript "SICNetseason V1.0: a transformer-based deep learning model for seasonal Arctic sea ice prediction by incorporating sea ice thickness data" by Ren et al presents a seasonal forecast model for Arctic sea ice, based on deep learning. This paper presents a novel approach to training DL models for sea ice prediction, by integrating a loss function which considers spatial information (the Integrated Ice Edge Error), as well as the standard Mean-Squred Error (MSE). The authors claim that, by including PIOMAS sea ice thickness reanalysis in their training, SICNet$_{season}$ is able to "optimize" the spring predictability barrier, improving forecasts of September sea ice made before June 1st. These forecasts are benchmarked against a damped persistence forecast, and the ECMWF SEAS5 seasonal prediction system. Overall the paper is well written and is a nice contribution to the sea ice prediction literature. I recommend minor revisions before publication.

**Response:** Thanks for the comment.

**General comments**

**Comment 1**: One small comment I have relates to how the Spring Predictability Barrier (SBP) is motivated and referenced throughout the manuscript. I suggest changing statements like "optimize the SPB" to "optimize predictions around the SPB", and remove statements such as "overcome the SPB" on L60 and elsewhere. I think it's important to make it clear to the reader that the SPB is an inherent characteristic of Arctic sea ice that cannot be overcome by better data, as it relates to how physical sea ice mass anomalies are locked in by ice-albedo feedbacks at the date of melt onset. Having thickness data before the SPB is of limited use because thickness anomalies in winter-spring are primarily driven by export (and moderated by negative ice growth feedbacks), hence these anomalies do not persist for long.

**Response:** Agreed and revised. We revised statements like "optimize the SPB" or "overcome the SPB" to "optimize predictions around the SPB."

**Comment 2**: Another comment relates to how SICNet$_{season}$ is trained and evaluated. I think in the sea ice prediction community we would probably consider this leave-one-out evaluation as "cheating", as you have optimized the weights of the network using future data. As you say on L148, for a testing year 2000, you train using data from 1979-1999 and 2001-2019. Meanwhile If you were really making this forecast in 2000, you would only have had access to data from 1979-1999. Your model therefore has a much better understanding of sea ice variability and trends than it should have in the year 2000. This ultimately makes me question how fair it is to compare this model to damped persistence, unless you computed the damped persistence forecast in a similar leave-one-out way? For example, for a damped persistence forecast in the year 2000, are the anomalies at the chosen lead time based on a linear trend climatology computed over the period 1979-1999, or 1979-2019? The same question for the anomaly standard deviation and correlations. In any case, I think what would be most preferable is if the damped persistence forecasts were generated using only past data, and SICNetseason is trained for each forecast year, using only past data. Otherwise, I feel the only forecast evaluations I can

consider "fair" are those over 2020-2023.

**Response:** Thanks for the comment. The main reason for using the leave-one-out strategy is to evaluate the model's performance in a long time series with limited samples, which reviewers of a previous submission suggested. The sample volume for seasonal scale predictions with monthly mean data is not large. So, some statistical models [1-2] adopted the "leave-one-out" cross-validation to maximize the sample volume while obtaining a multi-year evaluation. Especially for deep learning models, the sample volume is vital for model training. If we train the model using data from 1979 to 1999 for the 2000 evaluation, the volume of training samples will be reduced by half. When we use the leave-one-out strategy, we randomly shuffle all samples for each training epoch to eliminate the influence of trends. The sea ice trends from the past have been disrupted. In this instance, the model can not learn the long-term trend. Besides, the ACC we calculated is the detrended ACC. These measures eliminate the contribution of the long-term trend to the model skill. Therefore, we have to utilize the "leave-one-out" strategy and try our best to eliminate the influence of the sea ice trend.

We used the Anomaly Persistence baseline, not the Damped Anomaly Persistence. Sorry for the misleading statement. We have clarified this point in the revision; see following Comment 12. As referred to in Yuan's papers [1], Persistence prediction is calculated using the current anomaly plus the climatology at the target time to estimate the future state. The climatology is the mean state of 1979-2019, excluding the target year.

[1] Yuan, X., Chen, D., Li, C., Wang, L., and Wang, W.: Arctic sea ice seasonal prediction by a linear markov model, J Clim, 29, 8151–8173, https://doi.org/10.1175/JCLI-D-15-0858.s1, 2016.
[2] Wang, Y., Yuan, X., Bi, H., Bushuk, M., Liang, Y., Li, C., and Huang, H.: Reassessing seasonal sea ice predictability of the Pacific-Arctic sector using a Markov model, Cryosphere, 16, 1141–1156, https://doi.org/10.5194/tc-16-1141-2022, 2022.

**Comment 3**: Lastly, I have concerns about the comparison with ICENet. On L268-271 you describe how you have changed ICENet's training procedure and architecture to be similar to SICNet$_{season}$ to make it a fairer comparison. I actually feel like this is less fair to ICENet. The original ICENet architecture, loss function, and inputs were optimized for the task outlined in the Andersson 2021 paper, and changing these may result in sub-optimal predictions. Effectively, you're no longer using ICENet. I suggest in this section you make a fair comparison to the original (unchanged) ICENet model, or you change the labelling to say you're comparing SICNet$_{season}$ with an (ICENet-inspired) U-Net architecture.

**Response:** Thanks for the comment. Agreed. The original IceNet treated the sea ice prediction (regression task) as a classification task. Here, we implemented the same backbone as the original IceNet and changed the classification output layer to a regression one. We adopted the comment and revised the "IceNet" labeling as "U-Net (IceNet-inspired)."

**Minor comments**

**Comment 1** L11: suggest changing "predictions made later than May" to "predictions made later than the date of melt onset (roughly May)."
**Response:** Agreed and revised.

**Comment 2** L17: suggest stating explicitly that the ECMWF model is the SEAS5 model
**Response:** Agreed and revised.

**Comment 3** L30: instead of referencing Andersson et al., 2021 here, I would reference papers specifically focused on ice-free timing, like Jahn et al 2024 and Kim et al 2023.
**Response:** Agreed and revised.

**Comment 4** L31: suggest changing to "it may weaken the stratospheric polar vortex", as actually Blackport et al., 2019 suggests that it likely does not.
**Response:** Agreed and revised.

**Comment 5** L43: suggest changing "before or on May" to "before or at the timing of melt onset"
**Response:** Agreed and revised.

**Comment 6** L45/46: Actually many statistical and dynamical models do beat damped persistence on these timescales. See the recent review paper by Bushuk et al 2024.
**Response:** Agreed. We delete this sentence in the revision.

**Comment 7** L61: suggest clarifying what you mean here by "mainstream" sea ice prediction
**Response:** Thanks for the comment. We revised the sentence as "numerical models are widely used in operationally sea ice predicting."

**Comment 8** L76: Clarify that this is the SEAS5 model
**Response:** Agreed and revised.

**Comment 9** L84: Is there a reason you don't use the more up-to-date (version 2) NSIDC sea ice concentration data set? https://doi.org/10.5067/MPYG15WAA4WX
**Response:** Thanks for the comment. We used the version 2 data set and made a wrong citation. We replaced the old reference with the following new one in the revision.

[1] DiGirolamo, N., Parkinson, C. L., Cavalieri, D. J., Gloersen, P. & Zwally, H. J. (2022). Sea Ice Concentrations from Nimbus-7 SMMR and DMSP SSM/I-SSMIS Passive Microwave Data. (NSIDC-0051, Version 2). [Data Set]. Boulder, Colorado USA. NASA National Snow and Ice Data Center Distributed Active Archive Center.

**Comment 10** L89: suggest adding that PIOMAS generally overestimates thin ice and underestimates thick ice regions
**Response:** Agreed and revised.

**Comment 11** L97: Just to clarify, the inputs to the network are monthly-mean fields, and you are predicting monthly-mean fields? So a Lead 4 prediction of September-mean SIC is based on monthly-mean May data?
**Response:** Agreed and revised. We added "the inputs to the network are monthly-mean fields" in the revision. The input length of the monthly mean SIC/SIT is six/three. So, a lead four prediction of September-mean SIC is based on monthly-mean SIC/SIT of Dec.-May/Mar.-May. We explain more about the input factors and their lengths in the revised Section 3.1.

**Comment 12** L167: suggest clarifying that by "Persistence model" you mean "Damped Anomaly Persistence"
**Response:** Thanks for the comment. The "Persistence model" we used is "Anomaly Persistence," not the "Damped Anomaly Persistence." It is calculated, referred to in Yuan's

paper [1], using the current anomaly plus the climatology at the target time to estimate the future state. As Yuan's studies show, the ACCs of "Anomaly Persistence" and "Damped Anomaly Persistence" in subseasonal are very similar, so we used "Anomaly Persistence" in our study. We clarify this point in the revision:

The Persistence is the anomaly persistence model. It assumes the anomaly constant in time and estimates the target SIC values by adding the current anomaly to the climate mean state at the target time, widely adopted as a benchmark for sea ice prediction (Wang et al., 2016).

[1] Wang, L., Yuan, X., Ting, M., and Li, C.: Predicting summer arctic sea ice concentration intraseasonal variability using a vector autoregressive model, J Clim, 29, 1529–1543, https://doi.org/10.1175/JCLI-D-15-0313.1, 2016.

**Comment 13** L204: Can you speculate here whether the lead 1 and 2 predictions from ECWMF are better because of their good atmospheric initialization? Certainly in Bushuk et al 2024, ECMWF SEAS5 beats all other dynamical forecast systems for Jun 1 to Sep 1 initializations, possibly for this reason. Did you test including atmospheric variables in your training?

**Response:** Thanks for the comment. Yes, that may be a reason. Zampieri et al. (2018) revealed that the ECMWF outperforms the climatology and many dynamical models in predicting SIC 0-45 days [1]. Bushuk et al. (2024) also showed that the RMSE of SEAS5 is lower than that of most statical models in Agu./Sep. 1 initialization [2]. These results demonstrate that the atmospheric initialization of SEAS5 may provide performance in sub-seasonal scale prediction. We did not include atmospheric variables in this study because an ablation experiment with different variables requires a lot of work and 20 years of testing. We have another paper focusing on evaluating the contributions of atmospheric variables (SAT, SST, surface radiation, SLP, etc.), which is under revision now. We revised the L204 as follows:

When the lead month is one, the MAE of SEAS5 is slightly better than that of Persistence and SICNet$_{season}$, indicating that the SEAS5 model performs well in monthly predicting. This result may be due to the good atmospheric initialization in SEAS5, which beat many machine learning and dynamical models in sub-seasonal scale SIC prediction (Bushuk et al., 2024).

[1] Zampieri, L., Goessling, H. F., and Jung, T.: Bright Prospects for Arctic Sea Ice Prediction on Subseasonal Time Scales, Geophys Res Lett, 45, 9731–9738, https://doi.org/10.1029/2018GL079394, 2018.
[2] Bushuk et al. Predicting September Arctic Sea Ice: A Multimodel Seasonal Skill Comparison. BAMS (2024).

**Comment 14** L228/229: change "cycles" to "circles."
**Response:** Agreed and revised.

**Comment 15** L253: change "generalization ability" to "generalization."
**Response:** Agreed and revised.

**Comment 16** Figures: I think generally the figures throughout the manuscript are quite small and it's difficult to read the numbers in the ACC/BACC plots (especially when the manuscript is printed). Also some of the spatial maps like Figure 7 are very busy with many panels, and it's hard to distinguish between contour lines without really zooming in (also please choose a different color for contours other than red and green for color blind readers). In figure 7 I

suggest just showing one or two example lead months, so that the individual panels can be made bigger and easier to see.

**Response:** Agreed and revised. In the revision, we plotted the figures using a large font size. We deleted some panels and kept only nine in Figure 7, lead months 4-6 of 2020, 2021, and 2023. The panels in Figure 7 are easier to read than before. The green and red lines have also been replaced by cyan and orange. Some new figures are shown as follows:

[Figure]

**Figure 2. Detrend ACC of SIE, BACC of SIE, and their differences of Persistence, SEAS5, and SICNet_season from Jun. to Sep., averaged by 2000-2019. (a)-(c) Detrend ACC of three models. Two detrend SIE series (predicted and observed) calculate each value. (d)-(e) Detrend ACC differences between SICNet_season and Persistence/SEAS5. (f)-(h) BACC of three models. Each BACC is a mean**

value during 20 testing years. (i)-(j) BACC differences of SICNet$_{season}$ and Persistence/SEAS5. The black line indicates the SPB: a maximum decrease between two adjacent lead months. The red signifies a high/improvement in ACC/BACC, and the blue signifies a decrease.

[Figure]

**Figure 4. Detrended ACC of SICNet$_{season\_nosit}$ (a) and SICNet$_{season}$ (b). (c) ACC difference obtained by SICNet$_{season}$ minus SICNet$_{season\_nosit}$. BACC of SICNet$_{season\_nosit}$ (d) and SICNet$_{season}$ (e). (f) BACC difference like (c). The red signifies a high/improvement in ACC/BACC, and the blue signifies a decrease.**

[Figure]

**Figure 6. BACC of 2020-2023. (a) Persistence, (b) SEAS5, and (c) SICNet$_{season}$. Each value is a mean value of the four testing years. The horizontal axis represents the six lead months, and the vertical axis represents the target months, Jun. to Sep. The red signifies a high/improvement in ACC/BACC, and the blue signifies a decrease.**

[Figure]

**Figure 7. Predicted Sep. SIEs and their BACCs of 2020/2022/2023 in four to six months lead by Persistence, SEAS5, and SICNet_season. (a)-(c) 2020, (d)-(f) 2022, and (g)-(i) 2023.**

[Figure]

**Figure 8. Detrended ACC of IceNet (a) and SICNet_season (b). (c) ACC difference obtained by SICNet_season minus U-Net (IceNet-inspired). BACC of U-Net (IceNet-inspired) (d) and SICNet_season (e). (f) BACC difference like (c). The red signifies a high/improvement in ACC/BACC, and the blue signifies a decrease.**

[Figure]

**Figure 9. The predicted Sep. SIEs of U-Net (IceNet-inspired) and SICNet_season in six months' lead: (a) 2012, (b) 2017, (c) 2018, and (d) 2019.**

**References**

Jahn et al. Projections of an ice-free Arctic Ocean. *Nature Reviews Earth and Environment* (2024).

Kim et al. Observationally-constrained projections of an ice-free Arctic even under a low emissions scenario. *Nature Communications* (2023).

Bushuk et al. Predicting September Arctic Sea Ice: A Multimodel Seasonal Skill Comparison. *BAMS* (2024).

**Response:** Thanks. We cite these references in the revision.